# Joint Cancer Segmentation and PI-RADS Classification on Multiparametric MRI Using MiniSegCaps Network

**DOI:** 10.3390/diagnostics13040615

**Published:** 2023-02-07

**Authors:** Wenting Jiang, Yingying Lin, Varut Vardhanabhuti, Yanzhen Ming, Peng Cao

**Affiliations:** Department of Diagnostic Radiology, University of Hong Kong, Hong Kong SAR, China

**Keywords:** prostate cancer, PI-RADS classification, multi-parametric MRI, CapsuleNet, convolutional neural network

## Abstract

MRI is the primary imaging approach for diagnosing prostate cancer. Prostate Imaging
Reporting and Data System (PI-RADS) on multiparametric MRI (mpMRI) provides fundamental
MRI interpretation guidelines but suffers from inter-reader variability. Deep learning networks show
great promise in automatic lesion segmentation and classification, which help to ease the burden
on radiologists and reduce inter-reader variability. In this study, we proposed a novel multi-branch
network, MiniSegCaps, for prostate cancer segmentation and PI-RADS classification on mpMRI.
MiniSeg branch outputted the segmentation in conjunction with PI-RADS prediction, guided by the
attention map from the CapsuleNet. CapsuleNet branch exploited the relative spatial information of
prostate cancer to anatomical structures, such as the zonal location of the lesion, which also reduced
the sample size requirement in training due to its equivariance properties. In addition, a gated
recurrent unit (GRU) is adopted to exploit spatial knowledge across slices, improving through-plane
consistency. Based on the clinical reports, we established a prostate mpMRI database from 462 patients
paired with radiologically estimated annotations. MiniSegCaps was trained and evaluated with
fivefold cross-validation. On 93 testing cases, our model achieved a 0.712 dice coefficient on lesion
segmentation, 89.18% accuracy, and 92.52% sensitivity on PI-RADS classification (PI-RADS ≥ 4) in
patient-level evaluation, significantly outperforming existing methods. In addition, a graphical user
interface (GUI) integrated into the clinical workflow can automatically produce diagnosis reports
based on the results from MiniSegCaps.

## 1. Introduction

Prostate cancer is the second leading cause of cancer death in men, with an estimated 1,414,259 new deaths in 2020 worldwide [1]. Prostate MRI is widely used to diagnose clinically significant prostate cancer in biopsy-naïve patients. It has been shown that prebiopsy MRI is superior to systematic biopsy in prostate cancer detection [2]. Multiparametric magnetic resonance imaging (mpMRI), including T2-weighted, diffusion-weighted, and dynamic contrast-enhanced scans, plays an integral role in enhanced visualization of prostate cancer, improved biopsy targeting, and monitoring proactive signs of disease progression [2]. In addition, MRI-guided biopsy is also increasingly adopted clinically for risk assessment, replacing the conventional transrectal ultrasound-guided biopsy [3].

The demand for prostate MRI is growing due to mounting evidence and guideline recommendations, and radiologists are facing a substantial increase in referrals [4]. The prostate imaging reporting and data system (PI-RADS) provides fundamental guidelines for assessing prostate MRI by classifying lesions into risk significance as the score increases [5]. PI-RADSv2 has been shown to help radiologists and specialists detect high-grade prostate lesions with high sensitivity [6]. However, PI-RADS has been hindered by poor inter-reader and intra-reader agreement [7]. Furthermore, PI-RADS reporting requires substantial expert knowledge, and radiologists with less experience had more significant inter-reader variability in PI-RADS scoring [8].

With recent developments in deep learning, convolutional neural networks (CNNs) exceeded human performance in natural image analysis [9], especially in image classification [9] and segmentation [10]. Deep learning-based artificial intelligence (AI) algorithms have shown great promise in medical image segmentation, detection, and classification [11,12,13] in recent years.

Deep learning-based lesion detection and PI-RADS classification algorithms are needed for prostate MRI reporting in clinical practice. Recent studies have shown the feasibility of detecting prostate cancer on mpMRI, using deep neural networks [14,15,16,17]. Some networks distinguished prostate cancer from normal tissues and provided the likelihood of prostate cancer [18,19,20]. Some studies compared the performance of network predictions with radiologists’ PI-RADS reports [18,21]. However, the existing method for PI-RADS classification is semi-automated, as manual lesion contours must be inputted into the model [22]. Additionally, these CNNs require a large amount of annotated data and data augmentation, alleviating class imbalance. Few of them addressed lesion detection and classification tasks jointly in one network [23,24] taking spatial relation between prostate lesion and zonal area into consideration which might be important for PI-RADS classification. The recent study only achieved reasonable performance at a PI-RADS cutoff value ≥ 4 [24] with three independent networks for lesion detection and classification which was time-consuming and required more data to train multiple networks. Therefore, this study aimed to solve above-mentioned issues and develop a multi-task network to segment and predict PI-RADS category efficiently by exploiting spatial relation between the prostate lesion and zonal area using limited data.

Capsule Network (CapsNet) [25] helps mitigate data starvation in deep learning-based medical image analysis due to its promising equivariance properties, representing the spatial/viewpoint variability of an object in a capsule (i.e., vector) format [26]. The basic idea of CapsNet is to encode the part-whole relationships (e.g., scale, locations, orientations, brightnesses) between various entities, i.e., objects, and parts of an object, to achieve viewpoint equivariance [25]. Unlike CNN models that learn all partial features of an object, CapsNet understands the relationship between these features through weights that are dynamically computed in each forward pass [25]. This optimization mechanism, dynamic routing, allows the contribution of parts to the whole object to be weighted differently during training and inference [25]. CapsNet has been applied to medical image segmentation in recent studies [27,28], demonstrating the prominence of hybrid architecture between Capsule-based and traditional neural networks in medical image analysis [27].

For PI-RADS, the categorization depends on not only the dimension, edge morphology, and signal intensity, but also the positional relations of the lesion (e.g., extraprostatic extension/invasion) and its zonal location relative to the transition zone (TZ) and peripheral zone (PZ) [5]. Specifically, each lesion can be scored 1–5 on diffusion-weighted imaging (DWI) and T2-weighted (T2W) MRI, as well as by the presence or absence of dynamic contrast enhancement. The contribution of these scores to the overall PI-RADS assessment differs depending on whether the lesion is located in the TZ or PZ of the prostate. For the TZ lesion, the PI-RADS assessment is primarily determined by the T2W score and sometimes modified by the DWI score. For the PZ lesion, the PI-RADS assessment is determined mainly by the DWI score and modulated by the presence of dynamic contrast enhancement [5]. Such spatial relationships and other lesion properties (location, scale, dimension, etc.) can be encoded and represented by CapsNet in a single capsule vector, which is helpful for prostate cancer detection and classification.

Due to our relatively small database, the model in this study was based on a multi-task network with MiniSeg [29] as the backbone for prostate cancer segmentation and PI-RADS classification. Our model, named MiniSegCaps, is built upon a 2D CapsuleNet block, which considers positional relations between the lesion, TZ, and PZ, and 2D convolutional encoder and decoder. Previous studies for PI-RADS classification based on convolutional blocks mainly used one-hot encoding for multi-class classification [23,24]. Different classes were assumed to be equally distanced, ignoring that cancer is a progressive disease. Inspired by Gleason Score prediction in [29], we adopt the ordinal encoding for different PI-RADS categories to encode the lesion progression into vectors. Ordinal encoded vectors are not mutually orthogonal and can suggest the similarities and differences between PI-RADS categories compared with one-hot encoding [29]. In our work, MiniSegCaps encodes five labels, i.e., four PI-RADS categories plus a normal issue type, into ordinal encoded vectors, predicting the ordinal encoded vector for each pixel using mpMRI. Mini-SegCaps can predict lesion masks, the scope (i.e., ordinal encoding) of PI-RADS categories, benign prostatic hyperplasia (BPH), and lesion malignancy. Inspired by a recurrent fully convolutional network (RFCN) [30], a gated recurrent units (GRU) module [31] leveraging inter-slice spatial dependences is also integrated into MiniSegCaps, to exploit the spatial information across adjacent slices, which are represented by capsule vectors, from the entire volume.

The contributions of our work are summarized as follows. Firstly, we proposed a multi-task network to segment and classify prostate cancer on mpMRI jointly. MiniSegCaps inherits the merits of both the 2D CapsuleNet to exploit spatial information and the 2D convolutional blocks to learn better visual representation. Secondly, for MiniSegCaps, we adopted the ordinal encoding to characterize PI-RADS score and the GRU on capsules to exploit spatial knowledge across slices in the entire volume.

## 2. Materials and Methods

### 2.1. MRI Protocol

Approval from the institutional review board was obtained for this retrospective study, including 462 patients who underwent prebiopsy MRI and prostate biopsy for network training and evaluation. Patients with prior radiotherapy or hormonal therapy were excluded. All MR imaging was performed on 3 T and 1.5 T scanners (286 patients on 3 T, 157 patients on 1.5 T; both 3 T and 1.5 T MRI systems were from GE Healthcare) with the standardized clinical mpMRI protocol, including T2 weighted (T2w) and apparent diffusion coefficients (ADC). We used axial T2w turbo spin-echo (TSE) imaging and maps of the ADC using echo-planar imaging (EPI) DWI sequence. For T2w, the echo time (TE)/repetition time (TR) was 80/5400 ms on 3 T, and TE/TR = 105/7100 ms on 1.5 T, respectively. T2w on both 3 T and 1.5 T had parameters: FOV = 348 mm × 348 mm, matrix = 512 × 512, number slice = 40, in-plane resolution = 0.68 mm/pixel, and through-plane resolution = 5 mm/slice. For reduced-FOV DWI, TE/TR = 80/5900 ms on 3 T, and TE/TR = 84/7000 ms on 1.5 T. Other parameters were identical on the two scanners: FOV = 300 mm × 300 mm, matrix size = 256 × 256, in-plane resolution = 1.17 mm/pixel, slice thickness = 4 mm, and three b-values = 0/700/1400 s/mm${}^{2}$. The ADC maps were obtained using a linear least-squares curve fitting of pixels (in log scale) in the three b0 and diffusion-weighted images against their corresponding b values.

Inclusion criteria for this study were (1) MRI performed between February 2014 and April 2021, (2) men aged from 37 to 90, and (3) patients diagnosed with a consecutive clinical examination or participation in our active surveillance program. We established a cohort of 569 patients with mp-MRI, including T2w, diffusion-weighted, and dynamic contrast material-enhanced MRIs performed at our institution. In this study, we only used two mpMRI sequences including T2w and diffusion-weighted imaging (DWI) due to the limited function of dynamic contrast material-enhanced (DCE) imaging in mpMRIs in the update of PI-RADSv2 [5]. Apparent diffusion coefficient (ADC) maps used in this study were obtained from corresponding DWI data. For this study, we focused on the 494/569 patients with one or more detectable prostate cancer lesions classified by radiologists due to PI-RADS score according to PI-RADSv2 [5]. Exclusion criteria were a history of treatment for prostate cancer (including antihormonal therapy, radiation therapy, focal therapy, and prostatectomy) and incomplete MRI sequences (either missing T2w or ADC images). After excluding 32 patients based on the exclusion criteria, imaging data from 462 patients due to PI-RADS score ≥ 1 were used in this study; each case has both T2w and ADC images.

PI-RADS scores for study cases were obtained from MRI reports based on PI-RADS interpretation of multiparametric MRI performed by four board-certified radiologists during the clinical routine. In addition, one radiologist reviewed all scans and manually delineated the contours of lesions on T2w and ADC images to provide ground truths (GT) based on clinical reports and their accompanying sector map diagrams. Note that completely encapsulated nodules of PI-RADS score = 1 were also labeled and grouped into one category (PI-RADS 1/2) with those of PI-RADS score = 2. Statistical details of the labels in each category were summarized in the table of Section 3.1. As in Figure 1, zonal masks of prostates on T2 images were obtained using pretrained UNet with an average dice of 0.89 on two public datasets (PROSTATEX [32], NCI-ISBI-2013 [33]). Preprocessing operations include resampling, normalization, cropping to the prostate region based on obtained zonal masks, and registration between T2w and ADC images, etc. (Figure 1).

### 2.2. MiniSegCaps for Prostate Cancer Detection and PI-RADS Classification

In this work, we proposed a multi-task network, MiniSegCaps, which inherits the merits of both CapsNet and CNNs. MiniSegCaps is an end-to-end multi-class CNN to jointly segment prostate lesions and predict their PI-RADS categories. MiniSegCaps adopts MiniSeg [34] as the backbone due to its outstanding performance on a small database, following the Unet-like encoder–decoder architecture [35].

As shown in Figure 2, a multi-task network MiniSegCaps with two predictive branches was proposed for detecting and classifying lesions. A concatenation of T2W, ADC, and zonal mask creates the model’s three-channel inputs. The decoder was one output branch trained for predicting lesion masks and PI-RADS scoping categories under the supervision of ordinal encoded lesion and BPH masks. Another output branch using two capsule layers (caps-branch) was attached to the end of the encoder to produce specific malignancy, i.e., low-grade or high-grade lesion. Reconstructed feature maps from the caps branch were also integrated into the decoder as an attention map to improve the performance of the decoder branch.

#### 2.2.1. Capsule Predictive Branch

The encoder and decoder of MiniSeg [29] extract high-dimensional features from inputs and generate segmentation, respectively. The model takes T2W, ADC, and zonal mask as its three-channel input. It converts the image information to high-dimensional features using a set of convolutional blocks in the encoder that capsules can further process. The learned features from the encoder are reshaped into a grid of H × W × D capsules, each represented as a single 256-dimensional vector. The capsule predictive branch contains two convolutional capsule layers capable of encoding spatial information of objects in the capsule vector. The number of capsule types in the last convolutional capsule layer equals the number of categories in the segmentation, which can be further supervised by a margin loss [25]. This branch is designed to predict the binary high-grade/low-grade PI-RADS categories. Three fully connected (FC) layers followed by the last convolution-capsule layer are also adopted to reconstruct the input features of the capsule predictive branch as in previous work [25]. The reconstructed features are also integrated into the input of the decoder by multiplying with the encoder’s learning components to focus the decoder’s attention on features relevant to a PI-RADS category.

#### 2.2.2. Ordinal Encoding for PI-RADS Categories

We use the decoder of MiniSeg to produce lesion segmentation and PI-RADS scoping categories under the supervision of multi-channel GT masks with ordinal encoding. A conventional multi-class encoder converts each label into a one-hot vector and predicts the one-hot vector through the multi-class output [26]. However, one-hot encoding assumes that different labels are independent of each other, and thus, the cross-entropy loss penalizes misclassifications equally, which ignores the progressiveness of PI-RADS categories. Therefore, we adopt an ordinal encoding [29,36,37] for different PI-RADS categories to encode the progressive lesion commonalities into vectors. Specifically, each bit of an ordinal vector identifies a non-mutually exclusive condition; thus, the k-th bit indicates whether the label is from a category greater than or equal to k (Table 1). In our model, the GTs are encoded into 4-channel masks supervising the model to produce 4-channel segmentation masks. Ordinal encoding characterizes the relationships between different labels; commonalities and differences between labels are represented as shared and distinct bits in an ordinal vector, thus allowing the model to learn the commonalities of all lesions and the distinction between different PI-RADS categories simultaneously [29].

#### 2.2.3. Capsule Gated Recurrent Unit (CapsGRU) for Volumetric Information Integration

To achieve coherent lesion segmentation and prediction of PI-RADS across different slices in one volume, we also introduced a gated recurrent unit (GRU) to exploit spatial dependences across adjacent slices, leveraging inter-slice spatial correlation. The GRU is added to the capsule prediction branch between the encoder and decoder, taking the features learned by the first capsule layer as input. This approach differs from the conventional GRU-Unet method, as our CapsGRU applies to capsules, unlike conventional GRU on feature maps. Here, we denote $e_{s}$ as the output of the first capsule layer where *s* indicates the slice index, i.e., sϵ1,…,S. This output consists of (256 × 6 × 6) feature maps. A recurrent mechanism [30] is introduced to extract global features that capture the spatial changes observed when moving from the base to the apex of the prostate by mapping $e_{s}$ into a new set of features, $h_{s}=\Phi \left(h_{s-1},e_{s}\right)$, where Φ(•) is a non-linear function, and the size of $h_{s}$ is the same as the size of $e_{s}$. The feature maps learn by the CapsGRU module and are then flowed to the next capsule predictive layer to produce PI-RADS categories.

### 2.3. Baseline Methods

Baseline methods in our study include U-Net [35], attention U-Net [38], U-Net++ [39], SegNet [40], MiniSeg [34], FocalNet [29]. To compare with MiniSegCaps, all these models are supervised with ordinal encoding GT segmentations during training. To keep consistent with our network, we adopt the same backbone MiniSeg for FocalNet. For a fair comparison, the training and validation workflows in Figure 2, consisting of image preprocessing, intensity normalization and variation, and image augmentation procedures, are applied equally to all methods.

### 2.4. Model Training and Evaluation

#### 2.4.1. Loss Functions

Four kinds of loss functions, including dice loss, binary cross-entropy (BCE) loss, margin loss, and mean-squared error (MSE) loss, are adopted in our network. The model was supervised with GT segmentations and PI-RADS categories. The dice loss and BCE loss with the weight of 0.5 are applied at the decoder head with multi-channel GT segmentations. The margin loss is applied to the capsule predictive branch with GT PI-RADS categories. We also use masked MSE loss scaled down by 0.0005 in the total loss, as in previous work [25], for reconstructing the capsule predictive branch.

#### 2.4.2. Implementation Details

The total loss is minimized by stochastic gradient descent (Adam) with an initial learning rate of 0.002, decaying by 0.8 every 20 steps. The network is trained for 300 epochs with a batch size of 256. The MRI images used for patch-based training are cropped to the same size of 100 × 100 to incorporate most prostate regions and further decrease the receptive field of lesion detection. In addition to image normalization, common image augmentations, including image shifting, scaling, and flipping, are also applied during the training. The image augmentations are performed for each batch of training images, not the validation images.

#### 2.4.3. Cross-Validation

We train and validate our model using fivefold cross-validation. Each fold consists of 280 or 281 training cases and 70 or 69 cases for validation. Each case contains 9 to 14 slices, and each fold of training and validation sets has around 3300 and 840 slices, respectively.

#### 2.4.4. Statistical Analysis

To calculate the sensitivity, specificity, positive predictive value (PPV), negative predictive value (NPV), F1 score, and accuracy in detecting prostate cancer, we defined true or false positives/negatives for an index lesion at the per-patient level. For example, a true positive means that a reader (deep learning method or radiologist) has correctly detected prostate cancer at the same location and categorized the lesion with at least the PI-RADS score category. In addition, the sensitivities and specificities of radiologists were compared to those of the deep learning methods.

### 2.5. Structure Reporting Graphic User Interface (GUI)

To further aid radiologists in prostate cancer diagnosis in clinical practice, we also designed a graphical user interface (GUI) integrated into the overall workflow to produce diagnosis reports of prostate cancer automatically, which contains the predicted lesion mask, lesion visualization on T2 and ADC, predicted probability of each PI-RADS category, position, and the dimension of each lesion, etc. The main steps of the workflow in GUI include image data importation, zonal segmentation, lesion overlay on mpMRIs, image preprocessing (cropping, normalization, etc.), lesion segmentation, PI-RADS classification, and diagnostic report generation (as shown in Figure 3).

GUI components support image processing toolboxes, picture archiving integration, and integration of deep learning libraries. For clinical AI applications of prostate cancer diagnosis, GUI components included prostate cancer segmentation, PI-RADS classification, and quantification. GUI was deployed with Streamlit software (version 1.2.0, https://streamlit.io/, accessed on 1 May 2022), integrating image processing toolboxes for image preprocessing and postprocessing and training deep learning models for zonal segmentation, lesion segmentation, and PI-RADS classification.

From the MiniSeg branch, the ordinal encoding channel with the the largest number of non-zero pixels was chosen as the final cancer segmentation and PI-RADS prediction. In our generated diagnostic report, the lesion location, i.e., peripheral/transition zone and left/right, was obtained from the intersection between the predicted lesion mask and zonal mask and the relative position of the lesion to the midpoint of the image. The lesion dimensions were computed by the product of the image resolution and the number of lesion pixels.

## 3. Results

### 3.1. Study Sample Characteristics

As shown in Table 2, the study sample was randomly divided into a subset used for training and a separate test set. In the training sample, 365 men had 3196 lesions, including 441 of 3196 (14%) PI-RADS category 2, 895 of 3196 (28%) PI-RADS category 3, 1098 of 3196 (34%) PI-RADS category 4, and 798 of 3196 (24%) PI-RADS category 5 lesions. The 93 test patients had 1021 lesions, including 142 of 1021 (14%) PI-RADS category 2, 281 of 1021 (27%) PI-RADS category 3, 353 of 1021 (35%) PI-RADS category 4, and 245 of 1021 (24%) PI-RADS category 5 lesions.

### 3.2. Prostate Lesion Segmentation

We compare our MiniSegCaps with SOTA baseline segmentation approaches in prostate cancer segmentation evaluated on the testing cohort using the Dice coefficient metric. In addition, a combination version of MiniSeg and CapsuleNet, supervised by ordinal encoding GTs, was also implemented to prove the effectiveness of incorporating Capsule layers into MiniSegCaps. Table 3 and Table 4 show the image-level and patient-level evaluation of these algorithms on prostate cancer segmentation and benign nodule segmentation.

Among baseline methods, 2D U-Net, attention U-Net, and U-Net++ have an averaged Dice coefficient of 51% lower than MiniSeg (65%) in image-level evaluation, as well as similar performance in patient-level evaluation, which indicates that a lightweight model (MiniSeg) performs better in dealing with a small database. Both MiniSegCaps and MiniSegCaps without CapsGRU in Table 3 outperform MiniSeg, SegNet, and FocalNet by a large margin. It indicates that our MiniSegCaps integrating Capsule layers can better identify prostate cancer from normal tissues by learning the relative spatial information of prostate cancer to different anatomical structures. Moreover, our MiniSegCaps obtained better results than MiniSegCaps w/o CapsGRU. As expected, CapsGRU in MiniSegCaps captured the spatial information across adjacent slices, boosting the prostate cancer segmentation performance. Our proposed MiniSegCaps finally improved the performance of prostate cancer segmentation by an average of 5% in the Dice coefficient compared with MiniSeg.

Figure 4 illustrates a visual comparison of the cropped T2, cropped ADC, cropped zonal mask, lesion ground truth (yellow contour), and predicted lesion mask by MiniSegCaps (red contour) on T2 and ADC images among five different cases. Our model successfully produced satisfactory segmentation of prostate cancer and revealed the spatial relationship between the zonal mask, lesion on T2, and ADC, which might help lesion location and classification. We also obtained consistent segmentations across adjacent slices within one volume, as shown in Figure 5. The results indicate that CapsGRU in MiniSegCaps helped capture spatial information and achieve better segmentations across adjacent slices.

Moreover, the results of benign nodule segmentation shown in Table 4 indicate that our model achieved the best performance among these algorithms. Figure 6 also shows a visual comparison of benign nodule ground truth (green contour) and predicted segmentation (blue contour) by deep learning models on T2 and ADC images among different cases.

### 3.3. PI-RADS Category Prediction

The quantitative results of SOTA baseline methods and our methods in prostate cancer classification in terms of both image-level and patient-level performance in the testing cohort are shown in Table 5, Table 6, Table 7, Table 8, Table 9, Table 10, Table 11 and Table 12.

For PI-RADS scoping classification (Table 6, Table 8 and Table 10), the average accuracy of the three categories produced by baseline methods was 57% (PI-RADS ≥ 3), 63% (PI-RADS ≥ 4), 65% (PI-RADS = 5) in the patient-level evaluation, respectively, slightly higher than those results in the image-level evaluation. Due to the consideration of the spatial relationship between objects, both of the last two models integrated capsule layers in Table 4, Table 5, Table 6, Table 7, Table 8 and Table 9 outperform the baseline methods substantially, improving the accuracy of classes (PI-RADS ≥ 3, PI-RADS ≥ 4, PI-RADS = 5) by 15%, 21%, 8%, respectively, and improving the sensitivities of (PI-RADS ≥ 4, PI-RADS = 5) by 5% on average, compared to those of MiniSeg in the patient-level evaluation. Furthermore, our MiniSegCaps consisting of Conv encoder, DeConv decoder with fused feature inputs, and Capsule predictive branch with CapsGRU, perform better in all PI-RADS classes, especially in patient-level evaluation, than MiniSegCaps w/o CapsGRU, the combination version of MiniSeg and CapsuleNet, which contains Conv encoder, DeConv decoder with fused feature inputs and capsule predictive branch. The CapsGRU in MiniSegCaps also improved consistency across adjacent slices to improve the PI-RADS classification results as expected. As a result, our MiniSegCaps achieved the best performance in all PI-RADS scoping classes. MiniSegCaps also improved the accuracy of PI-RADS scoping classification by an average of 15% in patient-level evaluation compared with MiniSeg.

For binary high-grade/low-grade PI-RADS classification (Table 11 and Table 12), the patient-level accuracy and sensitivity of the class (PI-RADS = 4/5) produced by MiniSegCaps were 71.56% and 76.32%, respectively. In addition, the CapsGRU in MiniSegCaps further improved the overall performance of binary high-grade/low-grade lesion differentiation compared with MiniSegCaps w/o CapsGRU.

## 4. Discussion

In this study, we presented a pipeline for automatic diagnosis of prostate cancer and a novel deep learning method, MiniSegCaps, for prostate cancer segmentation and PI-RADS classification on mpMRI. Our MiniSegCaps, a multi-branch network consisting of capsule predictive layers, exploited the spatial relationships between objects on mpMRI, and a CapsGRU to utilize spatial information across adjacent slices, predicting lesion segmentation and PI-RADS classifications jointly. We trained and validated MiniSegCaps under fivefold cross-validation using 462 pre-operative mpMRIs with annotations of all MRI-visible prostate cancer lesions and benign nodules (e.g., BPH). Experimental results show that our MiniSegCaps outperformed all six CNN-based baseline methods in PI-RADS classification, especially for PI-RADS ≥ 4, which is of great value for diagnosing clinically significant prostate cancer. For lesion segmentation, MiniSegCaps also achieved better performance than baseline methods. We also deployed a structured report GUI integrated into the overall workflow to automatically produce a prostate MRI report, which contained the predicted lesion mask, lesion visualization on T2 and ADC, and predicted probability of each PI-RADS category, position, and the dimension of each lesion, etc.

Previous studies for prostate cancer diagnosis on mpMRI mainly focused on prostate cancer detection based on classic segmentation networks such as U-Net [14,15,16,17,19,20,21]. They showed reasonable performance in distinguishing prostate cancer from normal tissues [14,15,16,17,19,20,21]. However, few considered the relative spatial information of prostate cancer to different anatomical structures to help identify prostate cancer from other anatomical tissues. Furthermore, most of them require a large amount of annotated data which are costly and difficult to acquire in practice. Instead, our MiniSegCaps can integrate anatomical knowledge by exploiting the spatial relationship between objects learned by the capsule branch. It can also be practicable on a small database due to the promising equivariance properties of capsule layers modeling the spatial/viewpoint variability of an object in the image and a lightweight MiniSeg backbone. Furthermore, the CapsGRU in our model is designed to utilize spatial information across adjacent slices, which could reduce flaws in the 2D model when dealing with a series of slices in volume. The experimental results in this study demonstrate that our MiniSegCaps achieved the best performance on prostate cancer detection compared with those classic segmentation networks.

The previous algorithms for PI-RADS classification did not address prostate lesion detection and required manual lesion contouring by radiologists [22]. However, recent work [21,24] comparing prostate cancer detection and PI-RADS classification between deep learning methods and radiologists showed comparable performance between classic CNNs and radiologists. However, the trained models in these studies achieved prostate cancer detection in separate steps. They cannot predict specific lesion segmentations [21], and mainly achieved good performance at a PI-RADS cutoff value ≥ 4. Furthermore, those studies only achieved classification on PI-RADS 3, 4, and 5, not including PI-RADS 1/2. Instead, our multi-branch MiniSegCaps can simultaneously produce prostate cancer masks by the decoder branch and the binary high-grade/low-grade PI-RADS categories (PI-RADS 1/2/3, 4/5) by the Capsule-predictive branch in an end-to-end diagnosis pipeline. Our MiniSegCaps also considered the anatomical priors by learning spatial relationships between objects and achieved the best performance on PI-RADS scoping classification compared with those existing deep neural networks. FocalNet [29] was a multi-class segmentation network for prostate cancer detection and Gleason scores prediction on mpMRIs with focal loss and mutual finding loss. We also applied this method to our prostate cancer segmentation and PI-RADS classification tasks for comparison. The experimental results demonstrate that our MiniSegCaps outperformed the FocalNet in lesion segmentation and PI-RADS classification due to exploiting the spatial relationship between objects for PI-RADS prediction by capsule predictive layers in comparison with FocalNet. A more recent study [23] used 3D cascaded U-Net, residual network architecture for prostate cancer detection, and PI-RADS classification, which required more training data than the 2D networks. Although it achieved acceptable performance in lesion segmentation (0.359 dice coefficient), the overall performance in the PI-RADS classification was unsatisfactory (30.8% accuracy). We tried a residual network (ResNet18 [41]) for PI-RADS classification based on segmented lesions. According to our experiments, the trained model was always prone to over-fitting during the training process, and the corresponding results on the testing cohort were unsatisfactory (30.56% accuracy). Instead, our MiniSegCaps showed prominence in dealing with a small database and achieved better performance in both lesion segmentation (0.712 dice coefficient) and PI-RADS classification (71.56% accuracy) in patient-level evaluation.

Our MiniSegCaps achieved satisfactory prostate cancer detection and PI-RADS classification compared with state-of-the-art methods, especially for cutoff at PI-RADS 3, with clinical importance in disease management. Distinguishing PI-RADS 2 and 3 is challenging for radiologists due to the insignificant difference in hypointense and homogeneous signal intensity on mpMRI. Meanwhile, determining PI-RADS 2 and 3 plays a pivotal role in the differential diagnosis of prostate cancer as it directly affects the clinical decision-making, i.e., biopsy or not. Specifically, prostate cancer with PI-RADS ≥ 3 usually requires a further biopsy to assess lesion aggressiveness by giving a histologically assigned Gleason score (GS) in clinical practice [2]. Therefore, accurate differentiation of PI-RADS 2/3 based on mpMRI reduces unnecessary biopsies [3].

Our multi-task MiniSegCaps can jointly predict prostate cancer masks and PI-RADS categories and achieve satisfactory performance by exploiting spatial relationship between objects. The proposed framework could assist inexperienced readers or non-experts in providing consultations about the prostate cancer contour and PI-RADS categorical probabilities. However, there are also some limitations to this study: (1) We only achieved satisfactory results on PI-RADS scoping and binary high-grade/low-grade classification, not including specific PI-RADS categories due to the sample size limitation; (2) The performance of lesion segmentation and PI-RADS classification on low scores needs further improvement; (3) This study only conducted experiments on local data from one imaging center, not including data from multiple centers. These problems are expected to be solved by incorporating more cases of low PI-RADS categories and enlarging the database from multiple centers. Incorporating the public data from different sites into our database, e.g., those from PROSTATEx [32] and NIH cancer image archive [42], to enlarge the database for PI-RADS classification is currently underway in our lab. In the future, we plan to extend our work to multi-center datasets, further improving the performance of our model on multi-center datasets by incorporating meta learning [43] or few-shot learning [44].

## 5. Conclusions

Our MiniSegCaps predicted lesion segmentation and PI-RADS classification jointly. As a result, our model achieved the best performance in prostate cancer segmentation and PI-RADS classification compared with state-of-the-art methods, especially for PI-RADS ≥ 3, which was highly important in clinical decision-making.

## Figures and Tables

**Figure 1 diagnostics-13-00615-f001:**
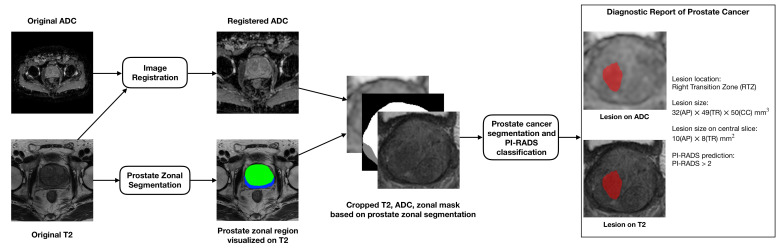
The overall pipeline of our work includes four main steps: (1) image preprocessing (registration, normalization, etc.); (2) zonal segmentation and cropping; (3) prostate cancer segmentation and classification; and (4) diagnostic report generation.

**Figure 2 diagnostics-13-00615-f002:**
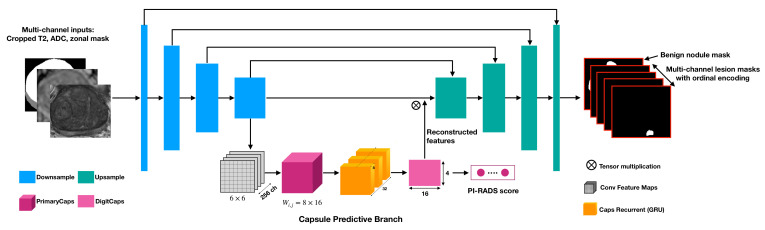
Architecture of the proposed MiniSegCaps, consisting of three parts: (1) MiniSeg, a lightweight segmentation network as the backbone for lesion mask prediction; (2) Capsule predictive branch for producing PI-RADS score; (3) CapsGRU module for exploiting spatial information across adjacent slices. The MiniSeg module extracts convolutional features maps from input mpMRIs and produces multi-channel masks for prostate cancer segmentation; learned features (6 × 6 × 256) by the last downsample block of MiniSeg are used as the inputs of capsule predictive branch for PI-RADS classification; with learned capsule feature stacks (8 × 32) by PrimaryCaps as inputs, the CapsGRU module exploits inter-slice spatial information during learning process; reconstructed features (6 × 6 × 256) by three fully connected layers [25] in the Capsule branch are also integrated into the MiniSeg module for better lesion identification.

**Figure 3 diagnostics-13-00615-f003:**
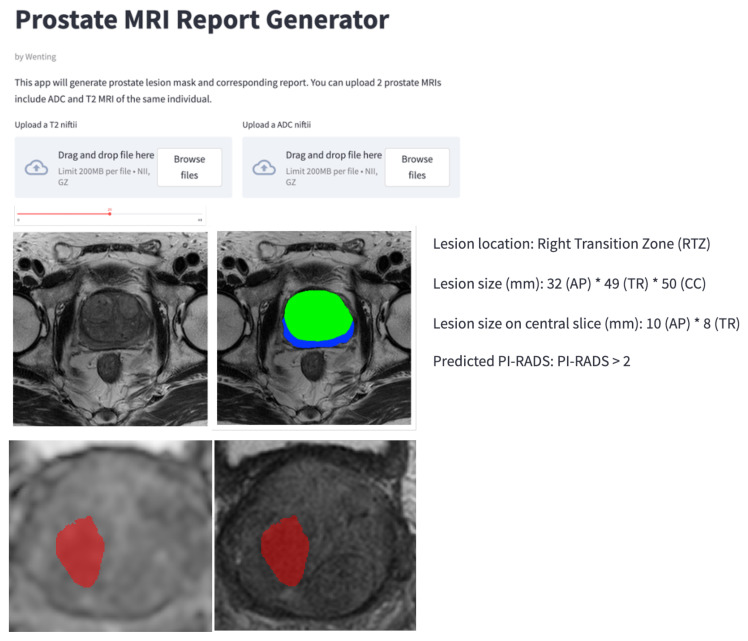
Graphical user interface (GUI) for automatic detection, PI-RADS classification, and diagnostic report generation of prostate cancer on mpMRIs.

**Figure 4 diagnostics-13-00615-f004:**
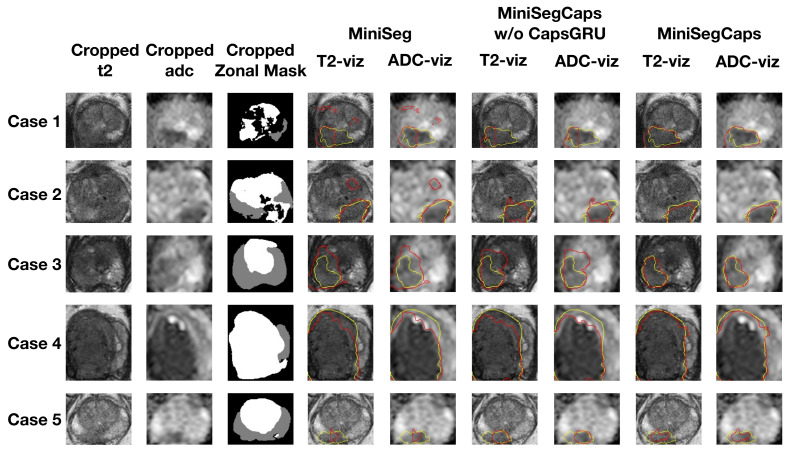
Visualization of lesion segmentation results among different cases. The yellow contour is the ground truth, and the red contours are from the deep learning predictions.

**Figure 5 diagnostics-13-00615-f005:**
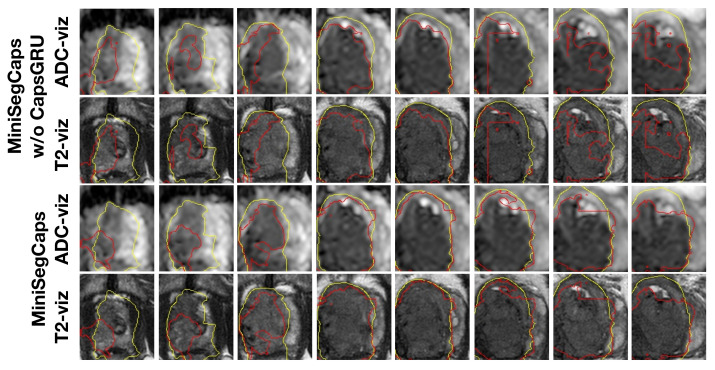
Visualization of lesion segmentation results on eight slices from one case. The yellow contour is ground truth, and the red contours are predictions from the MiniSegCaps without or with CapsGRU. MiniSegCaps with CapsGRU can better delineate the prostate cancer contours across different slices in one case compared to that without CapsGRU.

**Figure 6 diagnostics-13-00615-f006:**
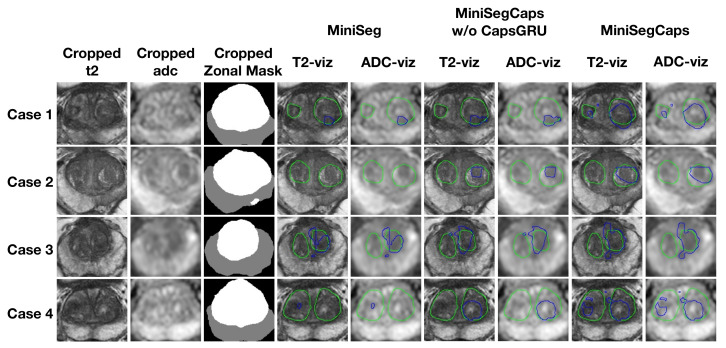
Visualization of benign nodule (e.g., BPH) segmentation results among different cases. The green contour is the ground truth, and the blue contours are from the deep learning predictions.

**Table 1 diagnostics-13-00615-t001:** PI-RADS category encoding for multi-class CNNs.

Label	Class	One-Hot Encoding	Ordinal Encoding
Non-lesion	0	1 0 0 0 0	0 0 0 0
PI-RADS 2	1	0 1 0 0 0	1 0 0 0
PI-RADS 3	2	0 0 1 0 0	1 1 0 0
PI-RADS 4	3	0 0 0 1 0	1 1 1 0
PI-RADS 5	4	0 0 0 0 1	1 1 1 1

**Table 2 diagnostics-13-00615-t002:** Characteristics and data splitting of the included study samples.

Database	MRI Scans	Total Lesions + Benign Nodules	PI-RADS 1/2 (Lesion + Benign Nodules)	PI-RADS 3	PI-RADS 4	PI-RADS 5
Train	365	626 + 719	68 + 719	150	246	162
Test	97	150 + 262	14 + 262	37	53	46
In Total	462	776 + 981	82 + 981	187	299	208

**Table 3 diagnostics-13-00615-t003:** Comparison of the image-level and patient-level performance between our method and baseline methods in prostate cancer segmentation (PI-RADS ≥ 3).

	Image Level	Patient Level
**Model**	**Dice Coefficient (DSC %)**	**Dice Coefficient (DSC %)**
2D U-Net	53.39	55.20
Attention U-Net	48.01	49.79
U-Net ++	49.57	51.06
FocalNet	62.59	64.38
SegNet	53.20	56.41
MiniSeg	64.89	66.73
MiniSegCaps w/o CapsGRU	68.26	70.89
MiniSegCaps	**70.17**	**71.16**

**Table 4 diagnostics-13-00615-t004:** Comparison of the image-level and patient-level performance between our method and baseline methods in benign nodule segmentation (e.g., BPH).

	Image Level	Patient Level
**Model**	**Dice Coefficient (DSC %)**	**Dice Coefficient (DSC %)**
2D U-Net	47.70	49.05
Attention U-Net	50.16	51.37
U-Net ++	46.26	48.03
FocalNet	55.01	56.59
SegNet	47.84	49.16
MiniSeg	55.63	56.74
MiniSegCaps w/o CapsGRU	56.35	57.98
MiniSegCaps	**57.96**	**59.14**

**Table 5 diagnostics-13-00615-t005:** Comparison of the image-level performance between our method and baseline methods in prostate cancer classification on PI-RADS ≥ 3.

	Image Level						
Model	Accuracy (%)	Sensitivity (%)	Specificity (%)	PPV (%)	NPV (%)	F1 Score (%)	Time per Image (s)
2D U-Net	64.13	29.23	97.09	90.48	59.22	44.19	0.03
Attention U-Net	56.05	85.81	33.42	49.50	75.60	62.78	0.03
U-Net ++	57.14	89.53	33.34	49.68	81.25	63.90	0.03
FocalNet	52.38	13.04	97.24	84.34	49.51	22.58	0.03
SegNet	53.81	75.53	37.98	47.02	68.06	57.96	0.03
MiniSeg	53.06	82.92	31.44	46.69	71.76	59.74	0.03
MiniSegCaps w/o CapsGRU	66.37	76.22	59.01	58.13	76.87	65.96	0.03
MiniSegCaps	**67.20**	77.37	59.20	59.89	76.87	**67.52**	0.03

**Table 6 diagnostics-13-00615-t006:** Comparison of the patient-level performance between our method and baseline methods in prostate cancer classification on PI-RADS ≥ 3.

	Patient Level						
Model	Accuracy (%)	Sensitivity (%)	Specificity (%)	PPV (%)	NPV (%)	F1 Score (%)	Time per Case (s)
2D U-Net	65.14	30.04	97.63	92.13	60.13	45.30	0.35
Attention U-Net	57.21	86.52	33.93	50.99	76.01	64.17	0.35
U-Net ++	56.05	87.68	32.73	49.02	78.26	62.88	0.35
FocalNet	51.98	12.52	96.62	80.72	49.41	21.68	0.35
SegNet	55.79	78.57	39.26	48.43	71.62	59.92	0.35
MiniSeg	54.67	84.62	32.52	48.13	74.07	61.35	0.35
MiniSegCaps w/o CapsGRU	68.80	79.44	60.84	60.28	79.82	68.55	0.35
MiniSegCaps	**70.20**	80.73	61.76	62.86	80.01	**70.68**	0.35

**Table 7 diagnostics-13-00615-t007:** Comparison of the image-level performance between our method and baseline methods in prostate cancer classification on PI-RADS ≥ 4.

	Image Level						
Model	Accuracy (%)	Sensitivity (%)	Specificity (%)	PPV (%)	NPV (%)	F1 Score (%)	Time per Image (s)
2D U-Net	67.35	52.23	85.07	80.39	60.32	63.32	0.03
Attention U-Net	57.85	88.29	32.06	52.41	76.36	65.77	0.03
U-Net ++	58.54	71.54	51.71	43.78	77.56	54.32	0.03
FocalNet	57.53	30.41	85.42	68.18	54.42	42.06	0.03
SegNet	63.70	80.82	44.55	62.77	65.85	70.66	0.03
MiniSeg	67.54	85.71	52.88	59.46	82.11	70.21	0.03
MiniSegCaps w/o CapsGRU	87.89	91.51	86.21	75.49	95.63	82.73	0.03
MiniSegCaps	**89.17**	**92.79**	**87.50**	77.44	**96.33**	**84.43**	0.03

**Table 8 diagnostics-13-00615-t008:** Comparison of the patient-level performance between our method and baseline methods in prostate cancer classification on PI-RADS ≥ 4.

	Patient Level						
Model	Accuracy (%)	Sensitivity (%)	Specificity (%)	PPV (%)	NPV (%)	F1 Score (%)	Time per Case (s)
2D U-Net	68.38	53.16	86.46	82.35	60.85	64.62	0.35
Attention U-Net	59.09	89.28	33.08	53.48	78.18	66.89	0.35
U-Net ++	59.66	73.17	52.56	44.78	78.85	55.56	0.35
FocalNet	58.22	31.33	86.62	71.21	54.42	43.52	0.35
SegNet	65.19	82.19	45.16	63.83	68.29	71.86	0.35
MiniSeg	68.41	86.45	53.68	60.36	82.93	71.09	0.35
MiniSegCaps w/o CapsGRU	88.07	91.67	86.41	75.86	95.69	83.01	0.35
MiniSegCaps	**89.18**	**92.52**	**87.58**	78.16	**96.06**	**84.74**	0.35

**Table 9 diagnostics-13-00615-t009:** Comparison of the image-level performance between our method and baseline methods in prostate cancer classification on PI-RADS = 5.

	Image Level						
Model	Accuracy (%)	Sensitivity (%)	Specificity (%)	PPV (%)	NPV (%)	F1 Score (%)	Time per Image (s)
2D U-Net	56.93	29.53	82.42	60.98	55.69	39.79	0.03
Attention U-Net	72.87	86.89	64.10	60.23	88.65	71.14	0.03
U-Net ++	75.39	86.89	68.21	63.10	89.26	73.10	0.03
FocalNet	56.92	26.31	84.62	60.75	55.93	36.72	0.03
SegNet	63.81	66.51	60.13	69.50	56.79	67.97	0.03
MiniSeg	79.07	76.74	79.88	57.14	90.76	65.51	0.03
MiniSegCaps w/o CapsGRU	86.94	80.41	89.18	71.89	92.98	75.91	0.03
MiniSegCaps	**87.34**	**81.44**	**89.36**	**72.48**	**93.33**	**76.70**	0.03

**Table 10 diagnostics-13-00615-t010:** Comparison of the patient-level performance between our method and baseline methods in prostate cancer classification on PI-RADS = 5.

	Patient Level						
Model	Accuracy (%)	Sensitivity (%)	Specificity (%)	PPV (%)	NPV (%)	F1 Score (%)	Time per Case (s)
2D U-Net	63.44	69.05	42.37	35.26	71.67	51.67	0.35
Attention U-Net	61.14	71.43	46.51	39.47	76.92	54.52	0.35
U-Net ++	62.18	69.75	51.78	33.93	70.06	48.13	0.35
FocalNet	58.33	38.10	74.07	53.33	60.61	44.43	0.35
SegNet	65.74	71.15	60.71	62.71	69.39	66.67	0.35
MiniSeg	80.72	79.17	81.36	63.33	90.57	70.37	0.35
MiniSegCaps w/o CapsGRU	87.25	81.82	89.86	79.41	91.18	80.60	0.35
MiniSegCaps	**88.24**	**84.85**	89.86	**80.00**	**92.54**	**82.35**	0.35

**Table 11 diagnostics-13-00615-t011:** Comparison of the image-level performance between our method and baseline methods in PI-RADS classification of prostate cancer.

	Image Level							
Model	PI-RADS	Accuracy (%)	Sensitivity (%)	Specificity (%)	PPV (%)	NPV (%)	F1 Score (%)	Time per Image (s)
MiniSegCaps w/o CapsGRU	1/2/3	64.75	57.96	76.97	81.91	50.44	67.88	0.03
MiniSegCaps	1/2/3	**69.64**	**66.61**	75.10	**82.80**	**55.56**	**73.83**	0.03
MiniSegCaps w/o CapsGRU	4/5	64.75	76.97	57.96	50.44	81.91	60.94	0.03
MiniSegCaps	4/5	**69.64**	75.10	**66.61**	**55.56**	**82.80**	**63.87**	0.03

**Table 12 diagnostics-13-00615-t012:** Comparison of the patient-level performance between our method and baseline methods in PI-RADS classification of prostate cancer.

	Patient Level							
Model	PI-RADS	Accuracy (%)	Sensitivity (%)	Specificity (%)	PPV (%)	NPV (%)	F1 Score (%)	Time per Case (s)
MiniSegCaps w/o CapsGRU	1/2/3	66.32	59.68	78.79	84.09	50.98	69.81	0.35
MiniSegCaps	1/2/3	**71.56**	**69.01**	76.32	**84.48**	**56.86**	**75.97**	0.35
MiniSegCaps w/o CapsGRU	4/5	66.32	78.79	59.68	50.98	84.09	62.65	0.35
MiniSegCaps	4/5	**71.56**	76.32	**69.01**	**56.86**	**84.48**	**65.17**	0.35

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
