# Peer review of "Joint Cancer Segmentation and PI-RADS Classification on Multiparametric MRI Using MiniSegCaps Network"

_diagnostics, 2023, doi:10.3390/diagnostics13040615_

Round 1

Reviewer 1 Report

The contributions of the paper are satisfactory. However, authors are advised to do the following modifications.

1. 

1   Highlight the limitations of existing works discussed in the literature

2. Add one paragraph to briefly discuss the more recent development of segmentation works for disease detection including the following works [1-3] to strengthen your literature section.  

[1]   “An Efficient Blood-Cell Segmentation for the Detection of Hematological Disorders”, IEEE Transactions on Cybernetics, 2021.

[2]   “A Systematic Review on Recent Advancements in Deep and Machine Learning based Detection and Classification of Acute Lymphoblastic Leukemia,” IEEE Access, 2022.

[3]   “A review of automated methods for the detection of sickle cell disease”, IEEE Reviews in Biomedical Engineering, vol. 13, pp. 309-324, 2020.

3.      Give performance analysis in terms of computational time.

4.      Discuss some weaknesses of the proposed method.

Reviewer 2 Report

Paper title: Joint Cancer Segmentation and PI-RADS Classification on Multiparametric MRI using MiniSegCaps Network

 There are some points that need to be further clarified:

1-       The motivation for the study should be further emphasized, particularly; the main advantages of the results in the paper comparing with others should be clearly demonstrated. 

2-       The limitations of the study are better suited for the discussion in a separate sub-section after the discussion on results.

3-       The example section needs to be further expanded and include some remarks to show the effectiveness and efficiency of the proposed method, compared with others. 

4-       Some remarks on the main results would be necessary and helpful. 

5-       The literature review should be extended.

Round 2

Reviewer 2 Report

The authors have satisfactorily addressed my comments and I may recommend the manuscript for publication.